# MET-AICE v1.0: an operational data-driven sea ice prediction system for the European Arctic

Cyril Palerme[1], Johannes Röhrs[2], Thomas Lavergne[2], Jozef Rusin[2], Are Frode Kvanum[1, 4], Atle Macdonald Sørensen[2], Arne Melsom[2], Julien Brajard[3], Martina Idžanović[2], Marina Durán Moro[2], and Malte Müller[1, 4]

[1]Development Centre for Weather Forecasting, Norwegian Meteorological Institute, Oslo, Norway
[2]Ocean Department, Norwegian Meteorological Institute, Oslo, Norway
[3]Nansen Environmental and Remote Sensing Center, Bergen, Norway
[4]Section for Meteorology and Oceanography, Department of Geosciences, University of Oslo, Oslo, Norway

**Correspondence:** Cyril Palerme (cyril.palerme@met.no)

**Abstract.** There is an increasing need for reliable short-term sea ice forecasts that can support maritime operations in polar regions. While numerous studies have shown the potential of machine learning for sea ice forecasting, there are currently only a few operational data-driven sea ice prediction systems. Here, we introduce MET-AICE, a prediction system providing sea ice concentration forecasts for the next 10 days in the European Arctic. To our knowledge, it is the first operational data-driven prediction system designed for short-term sea ice forecasting. MET-AICE has been trained to predict sea ice concentration observations from the Advanced Microwave Scanning Radiometer 2 (AMSR2) at 5 km resolution. After one year of operation, we show that MET-AICE considerably outperforms persistence of AMSR2 observations (errors about 30 % lower on average), as well as forecasts from several dynamical models such as TOPAZ5, Barents-2.5km and the European Centre for Medium-Range Weather Forecasts (ECMWF) Integrated Forecasting System.

## 1 Introduction

There are growing economic and geopolitical interests in the Arctic affecting various sectors such as shipping, tourism, fishing, and resource extraction (Stocker et al., 2020; Gunnarsson, 2021; Müller et al., 2023). This has resulted in a recent increase of the maritime traffic of about 7 % per year over the past decade (Müller et al., 2023). However, sea ice still remains a source of hazards, in particular due to the remoteness of the polar regions. In order to reduce the risk of accidents, seafarers going to ice infested waters are required to get sea ice information before their journeys by the International Code for Ships Operating in Polar Waters since 2017 (Polar Code; Deggim, 2018). The primary source of sea ice information used by seafarers navigating in the Arctic usually consists of sea ice charts produced by national ice services and satellite products (Wagner et al., 2020; Copeland et al., 2024). While the current conditions can be assessed using such observations (though some of these data might already be outdated in areas with fast changing conditions), they do not allow to anticipate the sea ice evolution during the next few days. For route planning, skillful sea ice forecasts could be used (Veland et al., 2021), but dynamical sea ice prediction systems are often not able to outperform persistence of sea ice concentration (SIC) observations for short lead times (Melsom

et al., 2019; Röhrs et al., 2023; Palerme et al., 2024; Kvanum et al., 2025). Furthermore, the need for accurate sea ice forecasts is also growing due to increasing sea ice motion caused by sea ice thinning (Tandon et al., 2018; Tschudi et al., 2020).

Several studies have recently shown that data-driven sea ice forecasting systems trained on satellite observations can be skillful (e.g. Grigoryev et al., 2022; Ren et al., 2022; Chen et al., 2023; Keller et al., 2023; Lin et al., 2023; Koo and Rahnemoonfar, 2024), and can provide more accurate forecasts than dynamical models (Andersson et al., 2021; Palerme et al., 2024; Kvanum et al., 2025; Lin et al., 2025). However, most operational sea ice prediction systems are still based on dynamical models (e. g. Smith et al., 2016; Barton et al., 2021; Williams et al., 2021; Ponsoni et al., 2023; Röhrs et al., 2023; Paquin et al., 2024), and there have only been a few based on machine learning approaches. The first operational data-driven sea ice prediction system developed has been IceNet (Andersson et al., 2021), which provides forecasts of the probability that SIC exceeds 15 % for the next 6 months. For lead times from 2 to 6 months, IceNet outperforms the forecasts from the European Centre for Medium-Range Weather Forecasts (ECMWF) SEAS5 seasonal prediction system (Johnson et al., 2019), while running 2000 times faster on a laptop than SEAS5 on a supercomputer. Furthermore, Palerme and Müller (2021) developed post-processed sea ice drift forecasts for lead times up to 10 days using machine learning that have been delivered on the commercial application IcySea from 2020 to 2024 (https://driftnoise.com/icysea.html, von Schuckmann et al., 2021).

Sea ice changes on short-time scales are primarily driven by the atmosphere, and in particular by the wind (Mohammadi-Aragh et al., 2018; Yu et al., 2020). Hence, it is crucial to include predictors from weather forecasts when developing data-driven sea ice prediction systems, as suggested by previous studies. Grigoryev et al. (2022) reported an improvement of 5 to 15 % when using forecasts of 2-meter temperature, surface pressure and wind from the National Centers for Environmental Prediction (NCEP) operational Global Forecast System (GFS). Moreover, Palerme et al. (2024) assessed an error reduction of 7.7 % when using ECMWF forecasts of 2-meter temperature and 10-meter wind in addition to sea ice predictors. In this paper, we introduce MET-AICE, a data-driven sea ice prediction system producing 10-day SIC forecasts with daily time steps in the European Arctic. The predictors used in MET-AICE are derived from SIC observations provided by the Advanced Microwave Scanning Radiometer 2 (AMSR2), ECMWF weather forecasts, and a land sea mask. The forecasts have been produced daily since March 2024 and are publicly available (see code and data availability section). In section 2, the datasets used in MET-AICE and for verification are described. Then, the deep learning prediction system and the verification methods are presented in section 3. MET-AICE is evaluated and compared to the forecasts from three dynamical models in section 4, followed by the discussion and conclusions in section 5.

## 2 Data

### 2.1 Datasets used for MET-AICE

MET-AICE has been trained to predict daily SIC observations at about 5 km resolution derived from AMSR2 data using a three-step algorithm called reSICCI3LF (Rusin et al., 2024). The first step combines the AMSR2 microwave imagery channels at 19 and 37 GHz to derive a SIC field at about 15 km resolution with relatively low uncertainties. It then computes a higher resolution SIC field (about 5 km) using the two 89 GHz channels. This higher-resolution SIC has larger uncertainties due to

**Table 1.** List of predictors used in MET-AICE.

| Source | Variable | Time |
|--------|----------|------|
| AMSR2 | SIC observations | Day preceding the forecast start date |
| AMSR2 | Land sea mask | Constant predictor |
| ECMWF | 2-meter temperature | Mean value between the forecast start date and the predicted lead time |
| ECMWF | 10-meter x wind component | Mean value between the forecast start date and the predicted lead time |
| ECMWF | 10-meter y wind component | Mean value between the forecast start date and the predicted lead time |

the atmosphere being less transparent at these higher frequencies. Finally, the high resolution details obtained from the 89 GHz channels are added to the coarser resolution field to produce a final SIC field at 5 km resolution with relatively low uncertainties. For more details about this algorithm and product, we refer to Rusin et al. (2024) and to Palerme et al. (2024) where this product was shown to be more accurate for the ice edge position than the OSI-408-a product from the Ocean and Sea Ice Satellite Application Facility (OSI SAF), also derived from AMSR2.

The AMSR2 SIC observations from the day preceding the forecast start date are used as a predictor in MET-AICE, and can be considered as the initial SIC conditions of the prediction system (table 1). The land grid points in these observations are considered as ice-free ocean (0 % SIC) since only valid values can be provided to the neural networks. MET-AICE also uses a land sea mask as a predictor, as well as ECMWF weather high-resolution forecasts (HRES, grid spacing of 9 km) in 3 predictors that are the 2-meter temperature and the 10-meter wind (x and y components on the MET-AICE grid). For the predictors from ECMWF weather forecasts, the mean values between the forecast start date and the predicted lead time are used. Due to this choice, 10 different models were developed for predicting the SIC evolution during the next 10 days. All the predictors are normalized using the statistics from the training dataset (values ranging between 0 and 1), and projected onto the MET-AICE grid (Lambert azimuthal equal area at 5 km resolution) using nearest neighbor interpolation before providing them to the neural networks. In Palerme et al. (2024) and Kvanum et al. (2025), the SIC trend from passive microwave observations during the 5 days preceding the forecast start date was used as a predictor, but it was shown to have a negligible impact on the predictions. Due to the limited importance of this predictor and the potential impact of missing data on the production of the forecasts, no SIC trend from satellite observations is used in MET-AICE.

## 2.2 Datasets used for verification

### 2.2.1 Observations

The AMSR2 SIC product from Rusin et al. (2024) is used as reference in this study since MET-AICE has been trained to predict these observations. However, passive microwave products sometimes indicate some spurious sea ice along the coasts due to the difference in brightness temperatures over open water and land, as well as the spatial resolution of several kilometers

(Lavergne et al., 2019; Kern et al., 2020). In order to avoid this issue, the oceanic grid points closer than 20 km from the coasts are not taken into account when the forecasts are evaluated using AMSR2 SIC observations as reference.

In addition, the ice charts produced by the ice service of the Norwegian Meteorological Institute (JCOMM Expert Team on sea ice, 2017; Copeland et al., 2024) are used as an independent dataset for verification. The ice charts, which represent sea ice classes of concentration, are manually drawn by ice analysts during week days using several types of observations. Synthetic aperture radar (SAR) observations are used where they are available due to their high spatial resolution. In other areas, the analysts prioritize visible and infrared data, whereas passive microwave observations are only used where no higher resolution

data are available. Therefore, land contamination is much less present in the ice charts than in passive microwave observations, and we decided to take into account all oceanic grid points when the ice charts are used as reference (no coastal grid points excluded). Furthermore, there are fewer ice chart data accessible for verification than AMSR2 observations because the ice charts are not produced during weekends and public holidays.

### 2.2.2 Dynamical models

SIC forecasts from three dynamical models are used as benchmarks for evaluating the performances of MET-AICE. The forecasts from ECMWF Integrated Forecasting System starting at 00:00 UTC (hereafter referred to as "ECMWF IFS") are evaluated in this study. They are produced on an octahedral reduced Gaussian grid (O1280) at about 9 km resolution and cover lead times up to 15 days. ECMWF IFS is a coupled numerical weather prediction system with the ocean component consisting of the NEMO ocean model and the LIM2 sea ice model which uses a viscous-plastic rheology (ECMWF, 2023, 2024). The

initial oceanic conditions are based on the Operational Sea Surface Temperature and Sea Ice Analysis (OSTIA) system (Donlon et al., 2012) as well as the OCEAN5 ocean analysis (Zuo et al., 2018). ECMWF IFS is constantly developed and forecasts from different model cycles are used in this study (https://www.ecmwf.int/en/forecasts/documentation-and-support/changes-ecmwf-model).

The TOPAZ5 Arctic Ocean system (hereafter referred to as "TOPAZ5") produces daily operational ocean and sea ice fore-

100 casts that are delivered by the Copernicus Marine Service (Ali et al., 2025). It is based on the Hybrid Coordinates Ocean Model (HYCOM, Bleck, 2002; Chassignet et al., 2006) version 2.2.98 coupled with the Los Alamos Sea Ice Model CICE version 5.1 (Hunke et al., 2017) and the ECOSMO-II biogeochemical model (Yumruktepe et al., 2022). Data assimilation is performed weekly using a 100-member ensemble Kalman filter (EnKF). The forecasts consist of 10 members covering lead times up to 10 days with hourly time steps and a spatial resolution of 6.25 km. The atmospheric forcing comes from ECMWF weather

forecasts, and the forecasts are available from September 2023.

The Barents-2.5km Ensemble Prediction System (hereafter referred to as "Barents-2.5") is a regional ocean-sea ice dynamical model producing hourly forecasts with lead times up to 96 hours (Röhrs et al., 2023). It consists of the Regional Ocean Modeling System (ROMS) version 3.7 (Shchepetkin and McWilliams, 2005) coupled with the Los Alamos Sea Ice Model (CICE) version 5.1 (Hunke et al., 2017). Barents-2.5 has 42 vertical layers and a spatial resolution of 2.5 km. The ensemble

consists of 24 ensemble members with daily assimilation cycles (AMSR2 SIC from Rusin et al. (2024), sea surface temperature, in-situ temperature and salinity observations) as well as updates of atmospheric files four times a day (six members are

respectively updated at model hours 00:00, 06:00, 12:00, and 18:00 UTC). Only the members produced at 00:00 UTC are used here for a consistent comparison with MET-AICE. In five of the six ensemble members produced at 00:00 UTC, the atmospheric forcing comes from ECMWF ensemble forecasts (ECMWF-ENS) at 9 km resolution. The remaining member is forced by the regional weather prediction system AROME-Arctic (Müller et al., 2017) developed at the Norwegian Meteorological Institute, which has the same domain and spatial resolution as Barents-2.5. We chose to compare MET-AICE to Barents-2.5 since the same AMSR2 SIC observations are used to initialize both prediction systems. An ensemble Kalman filter is used to assimilate these observations in Barents-2.5, though the model's ensemble is underdispersed (lack of ensemble spread, see Idžanović et al. (2023)), which indicates an overly high confidence in the forecast and thus limits the impact of the assimilation.

This study primarily focuses on evaluating the MET-AICE operational forecasts that are deterministic and have daily time steps. Therefore, the daily means of SIC fields from the dynamical models are evaluated. These datasets consist of the high-resolution ECMWF IFS (HRES) forecasts, the TOPAZ5 ensemble mean, the Barents-2.5 member forced by AROME-Arctic, and the mean of Barents-2.5 members forced by ECMWF-ENS forecasts. In addition, we also investigated producing probabilistic forecasts with MET-AICE in this study. For this experiment, MET-AICE forecasts were compared to the first 10 ensemble members from ECMWF IFS (ENS), the 10 TOPAZ5 ensemble members, and the 5 Barents-2.5 ensemble members forced by ECMWF-ENS produced at 00:00 UTC.

## 3  Methods

### 3.1  Deep learning prediction system

MET-AICE is based on convolutional neural networks with a U-Net architecture (Ronneberger et al., 2015), including residual connections (He et al., 2016) and spatial attention blocks (Oktay et al., 2018). This architecture (Fig. 1) was shown to outperform the original U-Net model for short-term sea ice prediction (Palerme et al., 2024). While the residual connections are used in every layer of the neural network, the spatial attention blocks are only present in the layers of the decoder. The residual connections are used to avoid vanishing issues and ease neural network training (He et al., 2016), whereas the spatial attention blocks are designed to identify and give more attention to relevant areas during training (Oktay et al., 2018). The MET-AICE grid (480 x 544 grid points) has a spatial resolution of 5 km, and the neural networks are composed of five downsampling and five upsampling operations. 32 convolutional kernels are used in the first layer, and then the number of convolutional kernels is doubled at every layer in the encoder (and divided by two in the decoder). The models were trained to minimize the mean squared error during 100 epochs with a batch size of 4. An initial learning rate of 0.005 was used with an Adam optimizer, which was then reduced by a factor of 2 every 25 epochs. In order to avoid overfitting, the model version with lowest validation loss was selected during training for each lead time. The models contain about 39 million parameters.

The deep learning models were trained using one forecast per week during the period 2013 - 2020 (about 400 forecasts for each lead time). While using weekly data decreases the size of the training dataset, this also reduces the overlap between the forecasts for long lead times, which improved the performances for lead times longer than 1 day during the tuning phase (probably due to less overfitting). Nevertheless, all the forecasts available (daily data) produced in 2021 were used for validation

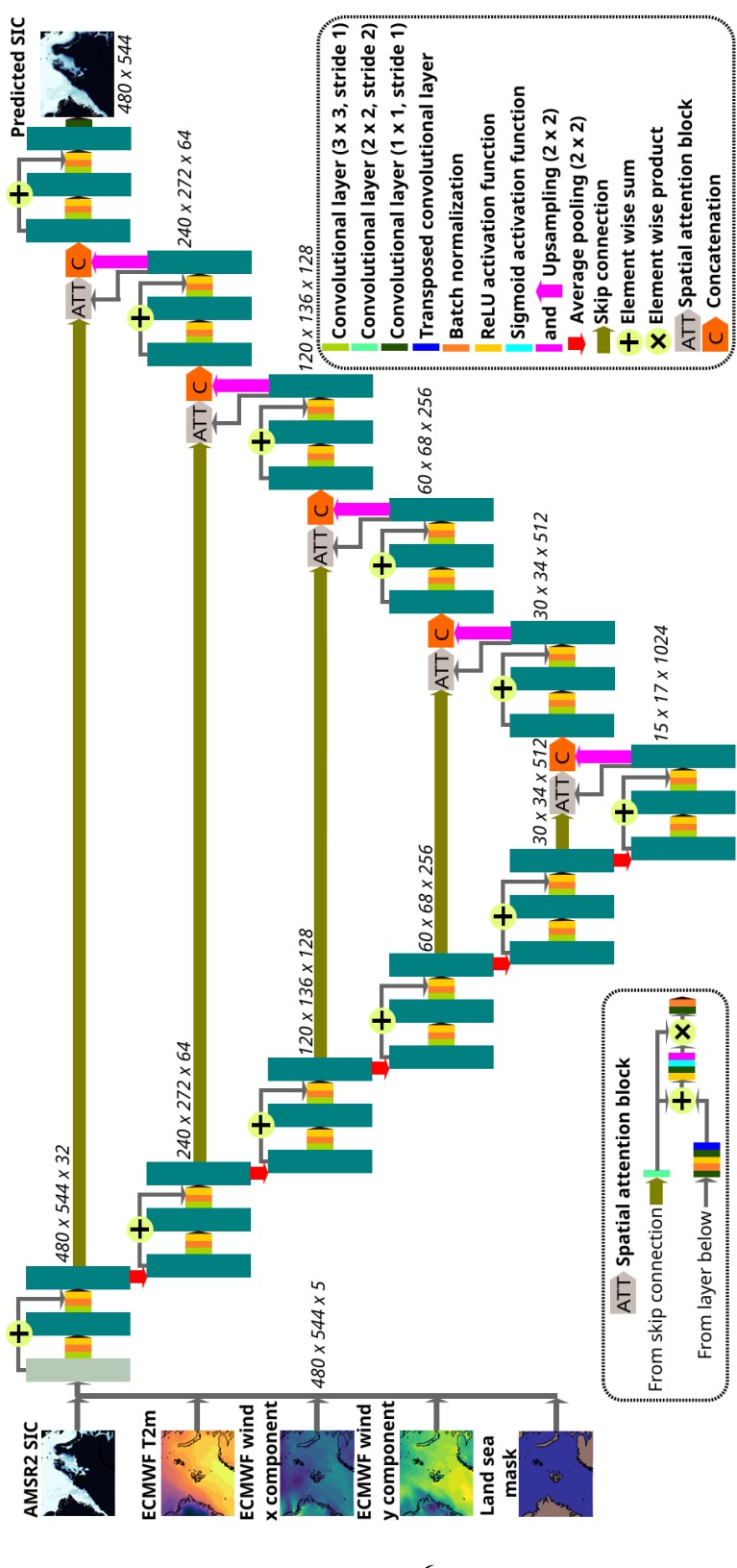

**Figure 1.** Architecture used in MET-AICE. The big green rectangles represent multi-channel feature maps. The dimensions of the feature maps (y, x, number of channels) are written next to each convolutional block.

(for tuning the models), and all the forecasts from 2022 were used for the test dataset during the development of MET-AICE. Furthermore, ECMWF weather forecasts produced by different model cycles were used for training the deep learning models due to the 8-year training period (https://www.ecmwf.int/en/forecasts/documentation-and-support/changes-ecmwf-model).

## 3.2  Verification

### 3.2.1  Verification scores

In this study, the SIC and the ice edge position (defined by the 10 % SIC contour and excluding coastlines) are evaluated. In order to assess the quality of SIC forecasts, the root mean square error (RMSE) of the SIC is calculated over all oceanic grid points using AMSR2 observations as reference. Since the ice charts do not provide SIC as a continuous variable but as sea ice classes, the ice charts are only used for evaluating the ice edge position. The ice edge position in deterministic forecasts is evaluated using the ratio of the Integrated Ice Edge Error (IIEE, Goessling et al., 2016) over the length of the ice edge in the

observations used as reference (hereafter referred to as "ice edge distance error", equation 1). This metric was introduced by Melsom et al. (2019) and provides the mean distance between the ice edges in the forecasts and in the observations. It is worth noting that normalizing the IIEE by the ice edge length (calculated using the method described in Melsom et al. (2019)) allows to use a metric which is not seasonally dependent, contrary to using the IIEE without normalization (Goessling et al., 2016; Palerme et al., 2019).

$$Ice\ edge\ distance\ error = \frac{IIEE}{Observed\ ice\ edge\ length} \tag{1}$$

While deterministic SIC forecasts are evaluated in most of this study, an experiment was performed with probabilistic forecasts. The ice edge position in probabilistic forecasts is assessed using the sea ice probability ($SIP_{forecasts}$ in equation 2), which is defined as the fraction of ensemble members with a SIC higher or equal to 10 %, whereas a binary sea ice probability is used for the observations ($SIP_{observations}$ in equation 2). Then, the forecasts are evaluated using the ratio of the Spatial Probability

Score (SPS, Goessling and Jung, 2018) over the ice edge length in the observations used as reference (hereafter referred to as "probabilistic ice edge distance error", equation 2). This metric was introduced by Palerme et al. (2019) and can be considered as the probabilistic extension of the ice edge distance error.

$$Probabilistic\ ice\ edge\ distance\ error = \frac{\int_x \int_y \left( SIP_{observations}(x,y) - SIP_{forecasts}(x,y) \right)^2 dydx}{Observed\ ice\ edge\ length} \tag{2}$$

### 3.2.2  Benchmark forecasts and evaluation period

In this study, the forecasts are considered skillful if they outperform persistence of AMSR2 SIC observations from the day before the forecast start date (hereafter referred to as "persistence of AMSR2 observations"). When the ice charts are used as reference, persistence of the ice charts from the day before the forecast start date (hereafter referred to as "persistence of ice

charts") is used as an additional benchmark forecast, though this reduces the number of days available for verification due to the lack of ice charts during weekends and public holidays.

In addition, we also developed a similar bias correction method for the dynamical models as the one used for TOPAZ4 in Palerme et al. (2024), where we calculate the difference between the first hourly time step from the dynamical models and the AMSR2 observations from the day before. Then, we subtract this difference from the forecasts for longer lead times in order to produce a benchmark forecast called "bias corrected forecasts" (the values lower than 0 % and higher than 100 % are then replaced by 0 % and 100 %, respectively). Due to the limitations of passive microwave SIC observations along the coasts

(where spurious sea ice can be reported), this bias correction was not applied to the grid points that were closer than 20 km from the coasts. Instead, the raw forecasts were used for areas close to land. This improves the impact of the bias correction when the forecasts are evaluated using the ice charts as reference, probably due to land contamination in the passive microwave observations. The bias correction used in this study is slightly different from the one proposed by Palerme et al. (2024) since the bias is calculated using the first hourly time step here whereas the first daily time step from TOPAZ4 was used in Palerme

et al. (2024), and because the bias correction was also applied to coastal grid points in Palerme et al. (2024).

    The forecasts are primarily evaluated during the period April 2024 - March 2025, which corresponds to the first year during which the MET-AICE forecasts have been produced operationally. During this period, the forecasts from the three dynamical models are available. While TOPAZ5 and ECMWF IFS cover the pan-Arctic, the domain of Barents-2.5 does not cover the full domain of MET-AICE (Fig.2 a). Due to the difference in model domains, the SIC forecasts are evaluated in the shared

domain between Barents-2.5 and MET-AICE after projecting all the datasets on the MET-AICE grid using nearest neighbor interpolation. It is also worth noting that a common land sea mask is used for all the datasets during verification (land if there is land in at least one of the dataset). In addition, we also evaluated MET-AICE and ECMWF IFS forecasts during the period 2022 - 2024 since only data from 2013 to 2021 were used during the development of MET-AICE (2013 - 2020 for training, and 2021 as validation dataset). TOPAZ5 and Barents-2.5 are not included in this comparison because TOPAZ5 forecasts have

only been available since September 2023, and Barents-2.5 forecasts have had lead times up to 4 days since December 2023. During the period 2022 - 2024, MET-AICE and ECMWF IFS forecasts are evaluated over the full MET-AICE domain after projecting all the datasets onto the MET-AICE grid.

## 4   Results

Fig. 2 shows the MET-AICE forecasts initialized on 01/02/2025 for lead times of 1, 4, and 10 days. During the first four
days of February 2025, the surface of a polynya located around Franz Josef Land grew before closing between 04/02/2025 and 10/02/2025. MET-AICE was able to reproduce relatively well the growth and the closing of this polynya, though the magnitude of the closing was underestimated. The retreat of the ice edge north of Svalbard during the first four days of February 2025 was also predicted successfully. There was also a decrease of the sea ice extent around Novaya Zemlya which was predicted by MET-AICE, though it was slightly underestimated. However, the sea ice patches located between Svalbard and Bjørnøya
remained longer than in MET-AICE predictions. Furthermore, the MET-AICE forecasts become smoother with increasing lead

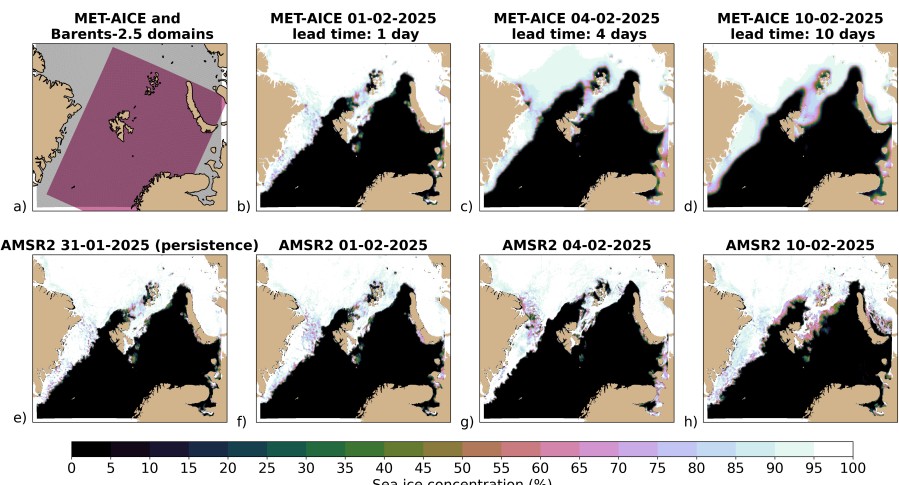

**Figure 2.** a) Domains of MET-AICE (gray) and Barents-2.5 (purple). Comparison between the MET-AICE forecasts initialized on the 01/02/2025 at 00:00 UTC (b to d) and the AMSR2 sea ice concentration observations for the corresponding days (f to h). e) The AMSR2 observations from the day preceding the forecast start date that are used as a predictor in the MET-AICE forecasts initialized on 01/02/2025.

times. This is due to the use of the mean squared error as loss function, as well as to the temporal averaging of the predictors from ECMWF weather forecasts.

In order to investigate to what extent the verification scores used in this study are influenced by the effective spatial resolution of the SIC fields, a sensitivity experiment was performed (Fig. 3). The AMSR2 SIC observations from 2013 to 2024 were used

to calculate the RMSE of the SIC and the ice edge distance error for 1-day persistence of AMSR2 retrievals. The observations from the first day were smoothed using various Gaussian filters with a standard deviation ranging from 5 km (1 grid point) to 30 km (6 grid points), and compared to the original AMSR2 SIC field from the second day. Only the ice edge length from the observations of the second day at the original spatial resolution (5 km) was used to compute the ice edge distance error regardless of the size of the Gaussian filter. In order to avoid some biases resulting from smoothing coastal grid points, the grid

points closer than 50 km from the coastlines were not taken into account in this analysis. Smoothing the AMSR2 SIC fields results in lower RMSE if the standard deviation of the Gaussian filter is between 5 and 20 km. The lowest RMSE is obtained for a Gaussian filter with a standard deviation of 10 km, with a reduction of 8.4 % compared to the RMSE obtained using the original SIC fields at 5 km resolution. This is in contrast with the ice edge distance error, which is only reduced by 0.5 % when a Gaussian filter with a standard deviation of 5 km is applied compared to the score obtained with the original resolution.

Moreover, the ice edge distance error constantly increases when larger Gaussian filters are applied. Therefore, the ice edge distance error does not really favor smoother SIC fields contrary to the RMSE of the SIC.

MET-AICE and ECMWF IFS forecasts are evaluated during the period 2022 - 2024 over the full domain of MET-AICE using AMSR2 observations as reference (Fig. 4). In addition, we evaluated the MET-AICE model developed for 1-day lead time in an auto-regressive mode to predict SIC up to 10 days ahead (hereafter referred to as "MET-AICE-roll-out"). In this

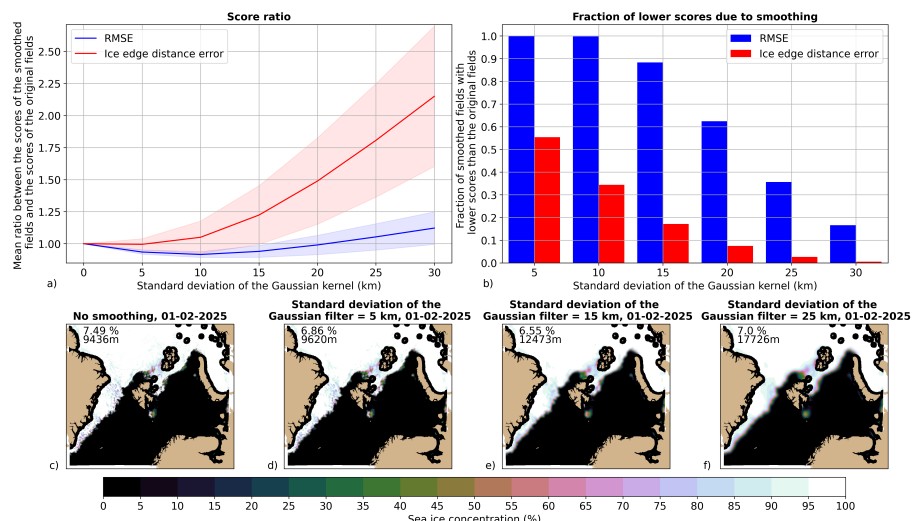

**Figure 3.** Influence of the effective spatial resolution on the verification scores. The AMSR2 observations during the period 2013 - 2024 were used to calculate the RMSE of the sea ice concentration and the ice edge distance error for 1-day persistence of AMSR2 observations over the MET-AICE domain (4327 days evaluated). The AMSR2 observations from the first day were smoothed using various Gaussian filters with a standard deviation ranging from 5 km (1 grid point) to 30 km (6 grid points), while the original AMSR2 observations from the next day were used as reference for computing the scores. The ice edge distance error was computed using the ice edge length from the observations of the second day at the original spatial resolution (5 km). The grid points closer than 50 km from the coastlines were excluded from this analysis. a) The mean ratio between the scores after smoothing and the scores using the original data (a score lower than 1 means that the smoothing results in a better score). The shaded areas represent the standard deviations of the verification scores. b) The fraction of scores that are better after smoothing the AMSR2 sea ice concentration fields. c) AMSR2 sea ice concentration observations at the original spatial resolution (5 km) on 01/02/2025. d, e, f) AMSR2 sea ice concentration observations on 01/02/2025 smoothed using Gaussian filters with a standard deviation of 5 km (d), 15 km (e), and 25 km (f). The scores on the top left corners of the maps show the RMSE of the sea ice concentration (top, in %) and the ice edge distance error (bottom, in meters) for the comparison between the AMSR2 observations from 01/02/2025 and 02/02/2025.

configuration, the SIC prediction from the previous lead time is used as a predictor to predict the next time step. While this configuration allows to improve the consistency between the time steps, it results in poorer performances than the operational version of MET-AICE. On average over all lead times, the RMSE of the SIC is about 19 % larger for MET-AICE-roll-out than for the operational version of MET-AICE, and the error for the ice edge position is about 10 % larger for MET-AICE-roll-out. Therefore, we decided to only present the results for the operational version of MET-AICE for the rest of this study. For all lead times and all the years evaluated, MET-AICE considerably outperforms persistence of AMSR2 observations (RMSE of the SIC and ice edge distance error about 28 % smaller on average), ECMWF IFS forecasts (RMSE of the SIC and ice edge distance error about 25 % and 30 % smaller, respectively), as well as ECMWF IFS bias corrected (RMSE of the SIC and ice edge distance error about 18 % and 21 % smaller, respectively). Furthermore, there is little interannual variability

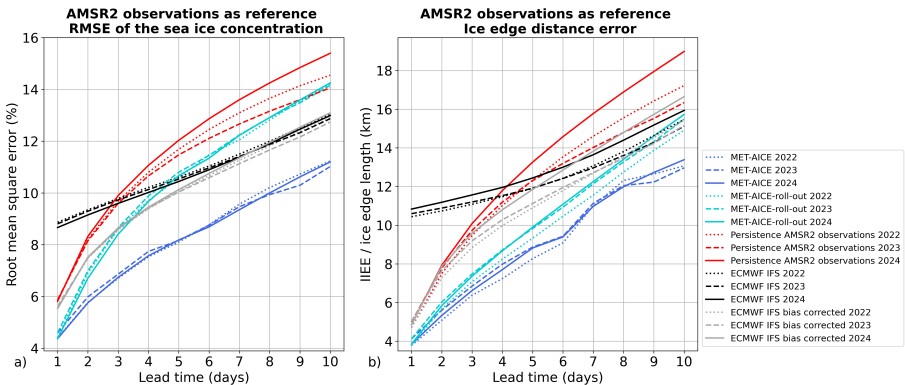

**Figure 4.** Performances of MET-AICE and ECMWF IFS during the years 2022, 2023, and 2024 over the full MET-AICE domain using AMSR2 observations as reference. MET-AICE-roll-out corresponds to using the MET-AICE model for 1-day lead time auto-regressively to predict the sea ice concentration for lead times up to 10 days. a) Root mean square error (RMSE) of the sea ice concentration. b) Evaluation of the ice edge position (defined by the 10 % sea ice concentration contour).

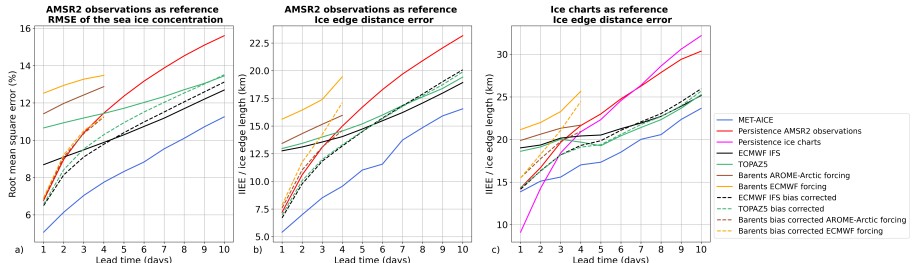

**Figure 5.** Performances of MET-AICE, ECMWF IFS, TOPAZ5 and Barents-2.5 during the period April 2024 - March 2025 over the shared domain between MET-AICE and Barents-2.5. a) Root mean square error (RMSE) of the sea ice concentration using AMSR2 observations as reference. b) Evaluation of the ice edge position (defined by the 10 % sea ice concentration contour) using AMSR2 observations as reference. c) Evaluation of the ice edge position using the ice charts as reference.

in the performances of MET-AICE, which suggests that re-training MET-AICE does not have to be done every year. While
ECMWF IFS does not outperform persistence of AMSR2 observations for lead times up to 3 days, the bias correction allows a considerable improvement of the forecast skill for lead times up to 6 days.

In Fig. 5, the performances of MET-AICE and the three dynamical models (ECMWF IFS, TOPAZ5, and Barents-2.5) during the period April 2024 - March 2025 are evaluated over the domain shared by MET-AICE and Barents-2.5. When the forecasts are evaluated using AMSR2 observations as reference (Fig. 5 a and b), MET-AICE outperforms all benchmark forecasts for all lead times. Compared to persistence of AMSR2 observations, MET-AICE has a RMSE of the SIC about 31 % smaller on average over all lead times (between 25 % and 33 % depending on lead time), and an ice edge distance error about 32 % smaller on average (between 25 % and 37 %). Nevertheless, MET-AICE is not able to outperform persistence of the ice charts for lead

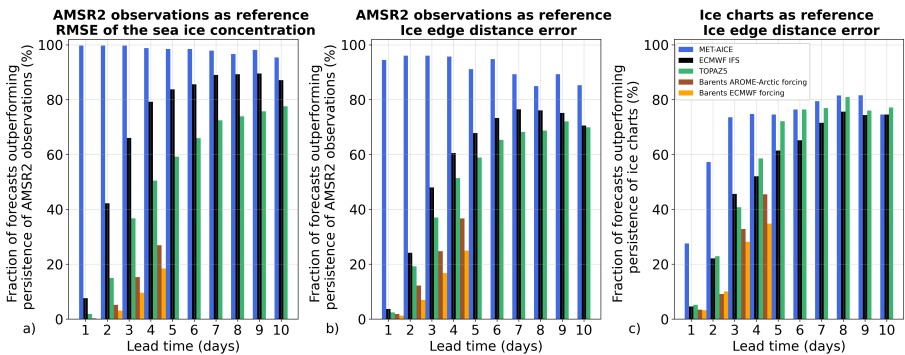

**Figure 6.** Fraction of days during which the forecasts outperform persistence of AMSR2 observations for the root mean square error of the sea ice concentration (a) and the ice edge position (b) using AMSR2 observations as reference during the period April 2024 - March 2025. c) Fraction of days during which the forecasts outperform persistence of the ice charts for the ice edge position using the ice charts as reference. The forecasts are evaluated over the shared domain between MET-AICE and Barents-2.5.

times of 1 and 2 days when the ice charts are used as reference due to the differences between the two observational products. However, MET-AICE has considerably lower errors for the ice edge position than all benchmark forecasts for longer lead times
(ice edge distance error about 23 % lower than for persistence of the ice charts for lead times from 3 to 10 days). Among the dynamical models, while ECMWF IFS has the best performances when AMSR2 observations are used as reference, TOPAZ5 has slightly better performances than ECMWF IFS when the forecasts are evaluated using ice charts as reference. It is also worth noting that Barents-2.5 performs considerably better when the model is forced by AROME-Arctic weather forecasts than when it is forced by ECMWF weather forecasts, which is consistent with the findings of Idžanović et al. (2023) regarding
surface currents.

    In Fig. 6, the fraction of days during which the forecasts can be considered skillful are shown. MET-AICE outperforms persistence of AMSR2 observations for the RMSE of the SIC between 95 % and 99.7 % of the days depending on lead time (98 % on average), and between 85 % and 96 % of the days (92 % on average) for the ice edge position using AMSR2 observations as reference. When the ice edge position is evaluated using the ice charts as reference, MET-AICE outperforms
persistence of the ice charts only 28 % of the days for 1-day lead time. Nevertheless, it outperforms persistence of the ice charts 57 % of the time for 2-day lead time, and at least 74 % of the days for longer lead times. While ECMWF IFS is the best dynamical model for all lead times when AMSR2 observations are used as reference, TOPAZ5 performs slightly better than ECMWF IFS when the ice charts are used as reference. Despite the assimilation of AMSR2 SIC observations, Barents-2.5 outperforms more often persistence of the ice charts using the ice charts as reference than persistence of AMSR2 observations
using AMSR2 observations as reference. This is consistent with Durán Moro et al. (2024) who also reported a better agreement between Barents-2.5 and the ice charts than with AMSR2 observations due to a higher SIC in the ice charts than in the AMSR2 product, and an over-generation of ice by the Barents-2.5 model (in particular in some areas such as the Fram Strait).

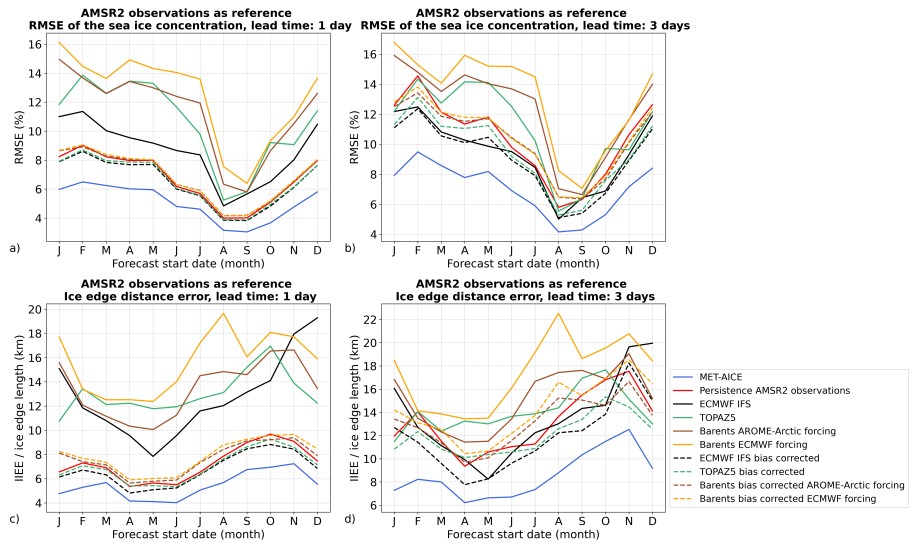

**Figure 7.** Seasonal variability of the performances of MET-AICE, ECMWF IFS, TOPAZ5, and Barents-2.5 for 1-day and 3-day lead times using AMSR2 observations as reference during the period April 2024 - March 2025. a, b) Root mean square error (RMSE) of the sea ice concentration. c, d) Evaluation of the ice edge position (defined by the 10 % sea ice concentration contour). The forecasts are evaluated over the shared domain between MET-AICE and Barents-2.5.

The seasonal variability in the performances of MET-AICE and the dynamical models for 1-day and 3-day lead times are shown in Fig. 7 and 8. When AMSR2 observations are used as reference (Fig. 7), MET-AICE considerably outperforms

persistence of AMSR2 observations for all the months for 1-day and 3-day lead times. This is in contrast with the dynamical models since none of them outperform persistence of AMSR2 observations for any month without bias correction for 1-day lead time. However, there is a much stronger seasonal variability when the ice charts are used as reference (Fig. 8) due to the differences between the two observational products (Palerme et al., 2024). For 1-day lead time, MET-AICE only slightly outperforms persistence of the ice charts in October, and has similar errors for the ice edge position as persistence of ice charts

in November and December. Nevertheless, MET-AICE outperforms persistence of the ice charts most of the year for 3-day lead time, but has larger errors than persistence of the ice charts between July and September.

Fig. 9 shows the spatial variability in the performances of MET-AICE when AMSR2 observations are used as reference. It is worth noting that this figure is the result of forecasts from different seasons, and that the location of the marginal ice zone varies considerably depending on the season. In order to keep only meaningful information, the grid points with fewer than 50

days during which sea ice is present (SIC higher than 0 %) between April 2024 and March 2025 are excluded from this figure. As expected, the largest errors occur where the marginal ice zone is often located, with the errors growing with increasing lead times. Furthermore, MET-AICE is able to outperform persistence of AMSR2 observations almost everywhere and for all lead times. The relative improvement compared to persistence of AMSR2 observations is larger for long lead times, in particular where the marginal ice zone is often located.

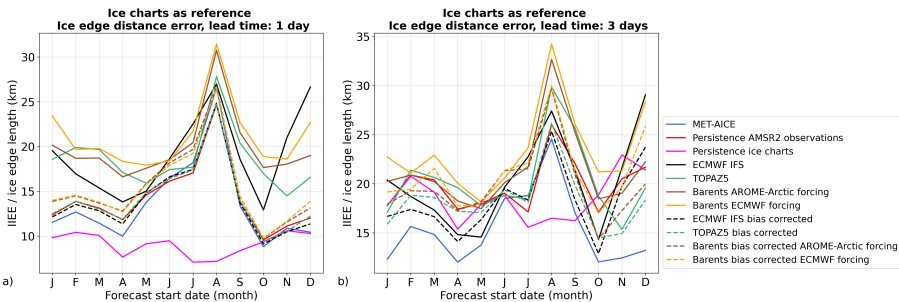

**Figure 8.** Seasonal variability of the performances of MET-AICE, ECMWF IFS, TOPAZ5, and Barents-2.5 for 1-day (a) and 3-day (b) lead times using the ice charts as reference during the period April 2024 - March 2025. The forecasts are evaluated over the shared domain between MET-AICE and Barents-2.5.

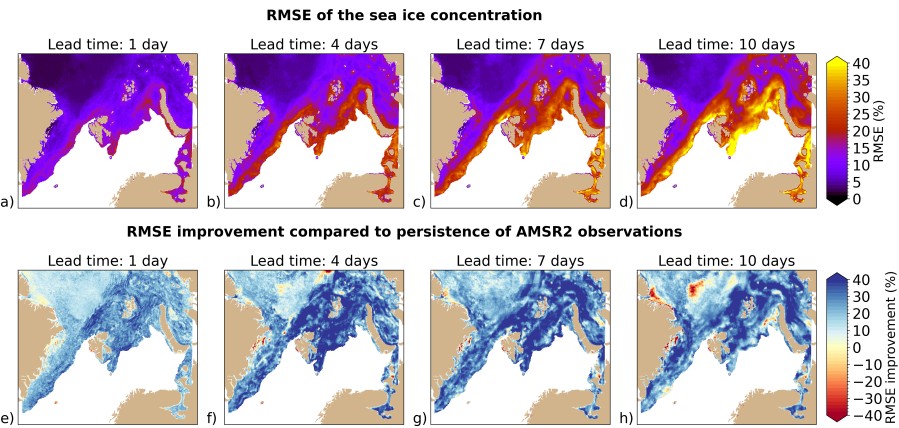

**Figure 9.** Spatial variability in the performances of MET-AICE using AMSR2 observations as reference. The top row shows the root mean square error (RMSE) of the sea ice concentration in MET-AICE forecasts during the period April 2024 - March 2025. The bottom row shows the relative improvement for the RMSE of the sea ice concentration of MET-AICE forecasts compared to persistence of AMSR2 observations. Positive values (blue) mean that MET-AICE outperforms persistence of AMSR2 observations. The grid points with less than 50 days during which AMSR2 observations indicate a sea ice concentration higher than 0 % are excluded from this analysis.

In addition to evaluating the operational MET-AICE deterministic forecasts, we investigated if we could use several ensemble members from ECMWF IFS (ECMWF-ENS) for producing probabilistic SIC forecasts with MET-AICE. We used the first 10 ensemble members of ECMWF-ENS for the atmospheric predictors (10-meter wind and 2-meter temperature) to produce a set of 10 ensemble members with MET-AICE. In Fig. 10, the forecasts starting on Mondays, Wednesdays, and Fridays from April 2024 to March 2025 are evaluated in the shared domain between MET-AICE and Barents-2.5 in order to include the three dynamical models. Overall, MET-AICE and the three dynamical models produce ensemble forecasts that are overconfident (not enough ensemble spread), which means that low sea ice probabilities (SIP) are observed more frequently than predicted whereas high SIP are observed less frequently than predicted. While the MET-AICE forecasts are particularly overconfident

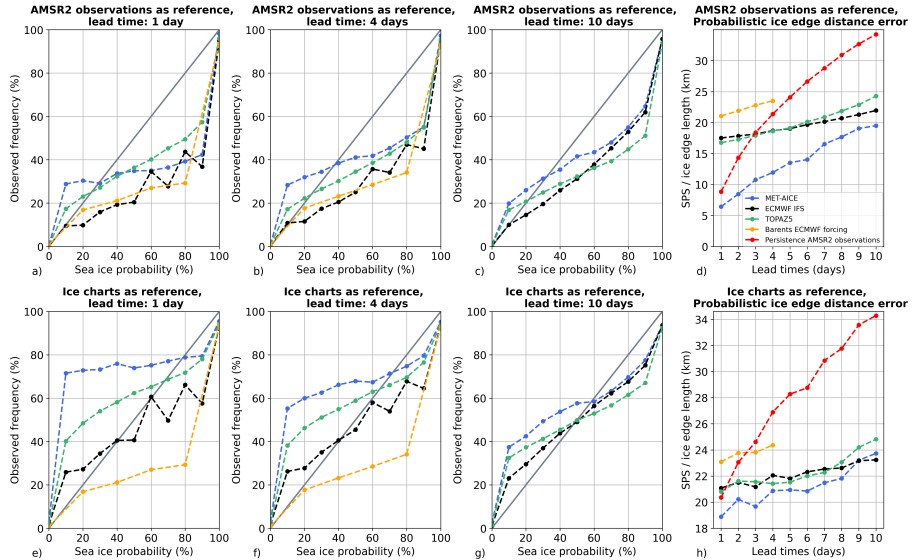

**Figure 10.** Evaluation of probabilistic sea ice forecasts. The forecasts starting on Mondays, Wednesdays, and Fridays from April 2024 to March 2025 are evaluated in the shared domain between MET-AICE and Barents-2.5. a, b, c) Comparison between the sea ice probability (probability that the sea ice concentration exceeds 10 %) and the observed frequency of occurrence in AMSR2 observations. d) Probabilistic ice edge distance error (ratio of the Spatial Probability Score (SPS) over the observed ice edge length) depending on the lead time using AMSR2 observations as reference. e, f, g) Comparison between the sea ice probability and the observed frequency of occurrence in the ice charts. h) Probabilistic ice edge distance error depending on the lead time using the ice charts as reference.

for 1-day lead time, they have a similar spread as the TOPAZ5 forecasts for 10-day lead time. Among the prediction systems, ECMWF IFS produces the largest ensemble spread. Furthermore, the probabilistic ice edge distance error is lower in MET-AICE forecasts than in all dynamical models, except for lead times of 9 and 10 days when the ice charts are used as reference.

## 5  Discussion and conclusion

While many studies have demonstrated that data-driven sea ice forecasts can be skillful and more accurate than dynamical models (e.g. Andersson et al., 2021; Grigoryev et al., 2022; Ren et al., 2022; Chen et al., 2023; Kvanum et al., 2025), there are currently only a few of these systems running operationally. To our knowledge, MET-AICE is the first operational data-driven sea ice forecasting system focused on short and intermediate time scales (1 to 10 days). The forecasts are produced daily and are publicly available (see code and data availability section). MET-AICE is skillful for predicting AMSR2 SIC observations (that were used for training the deep learning models), and produces more accurate forecasts than the dynamical models evaluated in this study (ECMWF IFS, TOPAZ5, and Barents-2.5), even after bias correction. It considerably outperforms persistence of AMSR2 observations for all lead times (the RMSE of the SIC is about 31 % lower on average, and the ice edge position is about 32 % more accurate on average).

Passive microwave SIC observations have some limitations such as melt pond ambiguity and land contamination (Kern et al., 2016, 2020; Lavergne et al., 2019). In particular, it was shown by Palerme et al. (2024) that the AMSR2 product used in MET-AICE has larger discrepancies with the ice charts close to the sea ice minimum (in August, September, and October) compared to the rest of the year. Therefore, MET-AICE has lower skill when the ice charts are used as reference and particularly close to the sea ice minimum. We also noticed that MET-AICE sometimes predicts too much sea ice along the coastlines, which is likely due to the land contamination of the passive microwave observations. It would be relevant to use other types of satellite SIC observations along the coasts in order to mitigate this issue in the future.

Whereas many dynamical sea ice models produce sea ice forecasts with hourly time steps (e.g. Williams et al., 2021; Ponsoni et al., 2023; Röhrs et al., 2023; Paquin et al., 2024), it is challenging to develop data-driven sea ice forecasts with such a high temporal resolution due to the limited number of observations available at this time scale. SIC and drift products with a pan-Arctic coverage are typically made available as daily averaged products (e.g. Tonboe et al., 2016; Lavergne et al., 2019; Tschudi et al., 2020; Wulf et al., 2024), limiting their usefulness for short lead times. One of the current limitations of dynamical sea ice models seems to be their inaccuracy at the initial state despite using data assimilation. For example, in the case of Barents-2.5, the large initial bias is likely due to the lack of ensemble spread, which mitigates the impact of data assimilation. A relatively simple bias correction at the initial state can greatly improve the accuracy of the forecasts as it was already shown by Palerme et al. (2024). Furthermore, this is also one of the main advantages of data-driven prediction systems which can have low biases if they are trained on a period long enough that captures the vast majority of the situations encountered.

The development of MET-AICE will continue in the future. It is planned to add predictions of sea ice drift and sea ice thickness since several studies have already shown the potential of machine learning for predicting these variables (e.g. Palerme and Müller, 2021; Hoffman et al., 2023; Durand et al., 2024; Koo and Rahnemoonfar, 2024; Zaatar et al., 2025). MET-AICE is currently a deterministic prediction system that was developed using the mean squared error as loss function. As a consequence, the forecasts become smoother with increasing lead times. In order to avoid this smoothing, generative artificial intelligence could be used to increase the resolution of the forecasts and create a set of ensemble members. For example, Finn et al. (2024) recently showed that diffusion models can be used to produce probabilistic sea ice forecasts without reducing the effective spatial resolution.

Furthermore, it is common practice to evaluate machine learning models using verification scores that are strongly correlated with the loss function. This study highlights that this can lead to spurious conclusions if no independent verification score is used in addition. Here, we show that the RMSE of the SIC can be reduced by 8.4 % due to the smoothing of the SIC fields. Therefore, we strongly recommend using more independent verification scores such as the ice edge distance error which does not favor smoother SIC fields as the RMSE of the SIC does.

*Code and data availability.* The codes used for developing and running the MET-AICE prediction system, as well as those used for the analysis are available through https://doi.org/10.5281/zenodo.17327120 (Palerme, 2025), and in the following GitHub repository (release v1.0.1): https://github.com/cyrilpalerme/MET-AICE-v1.0/. The MET-AICE forecasts, the AMSR2 sea ice concentration observations, and

the Barents-2.5 km EPS forecasts are available on the THREDDS server of the Norwegian Meteorological Institute (https://thredds.met.no/thredds/catalog/aice_files/catalog.html, https://thredds.met.no/thredds/catalog/cosi/AMSR2_SIC/catalog.html, and https://thredds.met.no/thredds/fou-hi/barents_eps.html). The TOPAZ5 forecasts are available through the Copernicus Marine Service (CMEMS) Marine Data Store (MDS): https://doi.org/10.48670/moi-00001. A license is needed to download the operational forecasts from the European Centre for Medium-Range Weather Forecasts (ECMWF).

*Author contributions.* CP: conceptualization, analysis, operationalization of the forecasts, writing (original draft), and funding acquisition. JR: Operationalization of the forecasts and writing. TL: conceptualization, analysis (remote sensing), writing, and funding acquisition. JR: analysis (remote sensing) and writing. AM: writing, and funding acquisition. JB: conceptualization and writing. AFK: conceptualization and writing. AMS: production of satellite observations. MI: analysis and writing. MDM: analysis and writing. MM: writing.

*Competing interests.* None of the authors has any competing interests.

*Acknowledgements.* The development of MET-AICE was supported by the SEAFARING project funded by the Norwegian Space Agency and the Copernicus Marine Service COSI project. The Copernicus Marine Service is implemented by Mercator Ocean in the framework of a delegation agreement with the European Union. The new AMSR2 SIC observations were developed with support from the SIRANO project (Research Council of Norway; grant no. 302917). We also thank the two anonymous reviewers for their comments which helped us to improve the manuscript.

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
