# Peer review of "MET-AICE v1.0: an operational data-driven sea ice prediction system for the European Arctic"

_EGUsphere, 2025_

## Author Comment (AC1)

Review of "MET-AICE v1.0: an operational data-driven sea ice prediction system for the European Arctic" by Palerme et al. Submitted to Geoscientific Model Development.

**General comments**

MET-AICE v1.0 is the first operational, data-driven sea ice prediction system specifically designed for short-term forecasts (1-10 days) in the European Arctic. The system is optimised for operational utility and higher spatial resolution, making it suitable for day-to-day maritime applications. The development of the MET-AICE system is particularly timely given the increasing demand for reliable, short-term, high-resolution sea ice forecasts, driven by increased maritime activity and heightened navigational risks associated with changing sea ice cover.

MET-AICE was trained on weekly AMSR2 weekly sea ice concentration data at 5-km resolution 2020 from the recently published reSICCI3LF algorithm, covering the period from 2013 to 2020. During training, the neural network models were iteratively updated over 100 epochs to minimize the mean squared error between the predicted SIC and the AMSR2 SIC observations. The system incorporates several predictors, including 9-km resolution ECMWF weather forecasts (2-m temperature and 10-m wind components), AMSR2 SIC observations from the day preceding the forecast start date, and a land-sea mask. MET-AICE uses a convolutional neural network with a U-Net architecture, designed specifically to capture spatial hierarchies in the input data. Operational forecasts have been generated since March 2024, with validation described in the manuscript covering a year-long period from April 2024 to March 2025. Despite demonstrated strengths in computational efficiency and accuracy compared to the Barents-2.5 km EPS model and other validation datasets, MET-AICE experiences reduced accuracy in coastal regions and diminished predictive skill during sea ice minimum periods, primarily related to inherent limitations in the input datasets. The current version of MET-AICE provides deterministic forecasts of sea ice concentration, which become smoother as the lead time increases. In future iterations, the authors plan to incorporate ensemble and probabilistic approaches to better quantify and represent the forecast uncertainty.

The paper is generally well written and structured, providing an important contribution towards operational high-resolution sea ice forecasting. However, several points need clarification before I can recommend the manuscript for publication.

I found the model description quite hard to follow. I wonder if you could include a flow diagram that shows the data inputs and preprocessing steps, a high level overview of the model architecture and key features (residual connections, spatial attention block and their purpose; downsampling and upsampling operations and progression of convolution kernels), and the outputs.

Thank you for this comment. We have added the flow diagram shown below in the revised version of the paper. Note that this figure is shown on an entire page in the revised version of

the paper. This describes what you mentioned. We agree that it helps to understand the architecture used in MET-AICE.

Figure 1. Architecture used in MET-AICE. The big green rectangles represent multi-channel feature maps. The dimensions of the feature maps (y, x, number of channels) are written next to each convolutional block.

The training period covers 7 years. Given the ongoing thinning and decline of sea ice cover, do you foresee a need for periodic retraining of the model? How might evolving sea ice conditions in the changing Arctic impact the model's forecasting accuracy over time?

We have added a new figure (see below) showing the interannual variability in the performances of MET-AICE and ECMWF IFS between 2022 and 2024. MET-AICE has very similar performances during these three years, suggesting that re-training the system is not necessary every year. We plan to re-train the models every time there will be modifications in the MET-AICE prediction system. We have added the following sentence in the results section:

"Furthermore, there is little interannual variability in the performances of MET-AICE, which suggests that re-training MET-AICE does not have to be done every year."

Figure 4. Performances of MET-AICE and ECMWF IFS during the years 2022, 2023, and 2024 over the full MET-AICE domain using AMSR2 observations as reference. MET-AICE-roll-out corresponds to using the MET-AICE model for 1-day lead time auto-regressively to predict the sea ice concentration for lead times up to 10 days. a) Root mean square error (RMSE) of the sea ice concentration. b) Evaluation of the ice edge position (defined by the 10 % sea ice concentration contour).

Training is based on weekly datasets, yet the forecasts are daily. I presume that using weekly training data enhances the model's generalization capability and robustness against short-term noise? However, this choice may limit the model's ability to capture rapid, short-term sea ice dynamics occurring at daily scales. How does this choice impact forecast accuracy during periods of rapid sea ice changes? Is the reduced forecast skill during sea ice minimum periods possibly related to a temporal limitation inherent in weekly training data?

MET-AICE is trained on daily AMSR2 sea ice concentration observations. Since this was not clear in the preprint, we modified the following sentence:

P2, line 50 "MET-AICE has been trained to predict SIC observations at about 5 km resolution derived from AMSR2 data using a three-step algorithm called reSICCI3LF (Rusin et al., 2024)."

by:

"MET-AICE has been trained to predict daily SIC observations at about 5 km resolution derived from AMSR2 data using a three-step algorithm called reSICCI3LF (Rusin et al., 2024)."

And the following sentence:

P5, line 114 "The deep learning models were trained using weekly data during the period 2013 - 2020 (about 400 forecasts for each lead time)"

was replaced by:

"The deep learning models were trained using one forecast per week during the period 2013 - 2020 (about 400 forecasts for each lead time)."

The evaluation spans a single year of operational forecasts. Although this period enables an analysis of seasonal performance and highlights the reduced skill during the summer, significant year-to-year variability in sea ice conditions may affect the robustness of the conclusions drawn. How confident are you in your findings after just one seasonal cycle, and could interannual variability impact where and when the model performs well? I am mostly thinking of how you might ultimately assign an uncertainty flag to the forecast data product.

We have added a new figure (see below) showing the interannual variability in the performances of MET-AICE and ECMWF IFS between 2022 and 2024. MET-AICE has very similar performances during these three years.

Figure 4. Performances of MET-AICE and ECMWF IFS during the years 2022, 2023, and 2024 over the full MET-AICE domain using AMSR2 observations as reference. MET-AICE-roll-out corresponds to using the MET-AICE model for 1-day lead time auto-regressively to predict the sea ice concentration for lead times up to 10 days. a) Root mean square error (RMSE) of the sea ice concentration. b) Evaluation of the ice edge position (defined by the 10 % sea ice concentration contour).

Furthermore we have added the following paragraph to describe this figure:

"MET-AICE and ECMWF IFS forecasts are evaluated during the period 2022 - 2024 over the full domain of MET-AICE using AMSR2 observations as reference (Fig. 4). In addition, we evaluated the MET-AICE model developed for 1-day lead time in an auto-regressive mode to

predict SIC up to 10 days ahead (hereafter referred to as "MET-AICE-roll-out"). In this configuration, the SIC prediction from the previous lead time is used as a predictor to predict the next time step. While this configuration allows to improve the consistency between the time steps, it results in poorer performances than the operational version of MET-AICE. On average over all lead times, the RMSE of the SIC is about 19 % larger for MET-AICE-roll-out than for the operational version of MET-AICE, and the error for the ice edge position is about 10 % larger for MET-AICE-roll-out. Therefore, we decided to only present the results for the operational version of MET-AICE for the rest of this study. For all lead times and all the years evaluated, MET-AICE considerably outperforms persistence of AMSR2 observations (RMSE of the SIC and ice edge distance error about 28 % smaller on average). ECMWF IFS forecasts (RMSE of the SIC and ice edge distance error about 25 % and 30 % smaller, respectively), as well as ECMWF IFS bias corrected (RMSE of the SIC and ice edge distance error about 18 % and 21 % smaller, respectively). Furthermore, there is little interannual variability in the performances of MET-AICE, which suggests that re-training MET-AICE does not have to be done every year. While ECMWF IFS does not outperform persistence of AMSR2 observations for lead times up to 3 days, the bias correction allows a considerable improvement of the forecast skill for lead times up to 6 days"

**The authors compare MET-AICE primarily to a single dynamical model, the Barents-2.5 km EPS. How does the performance of this dynamical model compare to other available dynamical models?**

We have added a comparison with the dynamical models TOPAZ5 distributed by the Copernicus Marine Service and the ECMWF Integrated Forecasting System in the revised version of the paper in order to provide more context. As an example, the figure below (which is in the revised version of the paper) shows the performances of MET-AICE compared to Barents-2.5km, ECMWF IFS, and TOPAZ5.

Figure 5. Performances of MET-AICE, ECMWF IFS, TOPAZ5 and Barents-2.5 during the period April 2024 - March 2025 over the shared domain between MET-AICE and Barents-2.5. a) Root mean square error (RMSE) of the sea ice concentration using AMSR2 observations as reference. b) Evaluation of the ice edge position (defined by the 10 % sea ice concentration contour) using AMSR2 observations as reference. c) Evaluation of the ice edge position using the ice charts as reference.

**Specific comments**

Line 63: It seems sensible to use 2-m temperature and 10-m winds to drive the system and you mention in the introduction that sea ice changes on short-time scales are driven by the wind. But was there any assessment of the optimal variables to train and run the model? At the very least it would be helpful to include references to justify your use of these variables to drive sea ice variability.

We think that this choice was already justified in the introduction of the preprint in lines 36 to 41:

"Sea ice changes on short-time scales are primarily driven by the atmosphere, and in particular by the wind (Mohammadi-Aragh et al., 2018; Yu et al., 2020). Hence, it is crucial to include predictors from weather forecasts when developing data-driven sea ice prediction systems, as suggested by previous studies. Grigoryev et al. (2022) reported an improvement of 5 to 15 % when forecasts from the National Centers for Environmental Prediction (NCEP) operational Global Forecast System (GFS) are used, whereas Palerme et al. (2024) assessed an error reduction of 7.7 % when using ECMWF weather forecasts in addition to sea ice predictors."

Nevertheless, we modified this paragraph with adding details about the atmospheric variables used in the studies from Grigoryev et al. (2022) and Palerme et al., 2024. The new paragraph is:

"Sea ice changes on short-time scales are primarily driven by the atmosphere, and in particular by the wind (Mohammadi-Aragh et al., 2018; Yu et al., 2020). Hence, it is crucial to include predictors from weather forecasts when developing data-driven sea ice prediction systems, as suggested by previous studies. Grigoryev et al. (2022) reported an improvement of 5 to 15 % when using forecasts of 2-meter temperature, surface pressure and wind from the National Centers for Environmental Prediction (NCEP) operational Global Forecast System (GFS). Moreover, Palerme et al. (2024) assessed an error reduction of 7.7 % when using ECMWF forecasts of 2-meter temperature and 10-meter wind in addition to sea ice predictors."

Line 65: I don't understand how the 10 different models were developed. Are each of these models for the different lead times, i.e. a set of 10 distinct forecasts for lead times of 1 day, 2 days, 3 days, all the way up to 10 days? Could you clarify the description here? Also, why do you have these different lead times - was the aim to find an appropriate lead time? Which is the dataset released via THREDDS? Is this the daily forecast with a 10-day lead time?

Yes, you are right. We developed a set of 10 distinct forecasts for lead times from 1 to 10 days. In the revised version of the paper, we added a comparison with a roll-out approach, in which the model for 1-day lead time is used to predict longer lead times auto-regressively (figure below). In this configuration, the sea ice concentration prediction from the previous lead time is used as a predictor to predict the next time step. The following sentences have been added to describe this comparison:

"In addition, we evaluated the MET-AICE model developed for 1-day lead time in an auto-regressive mode to predict SIC up to 10 days ahead (hereafter referred to as "MET-AICE-roll-out"). In this configuration, the SIC prediction from the previous lead time is used as a predictor to predict the next time step. While this configuration allows to improve the consistency between the time steps, this results in poorer performances than the operational version of MET-AICE. On average over all lead times, the RMSE of the SIC is about 19 % larger for MET-AICE-roll-out than for the operational version of MET-AICE, and the error for the ice edge position is about 10 % larger for MET-AICE-roll-out. Therefore, we decided to only present the results for the operational version of MET-AICE for the rest of this study."

Figure 4. Performances of MET-AICE and ECMWF IFS during the years 2022, 2023, and 2024 over the full MET-AICE domain using AMSR2 observations as reference. MET-AICE-roll-out corresponds to using the MET-AICE model for 1-day lead time auto-regressively to predict the sea ice concentration for lead times up to 10 days. a) Root mean square error (RMSE) of the sea ice concentration. b) Evaluation of the ice edge position (defined by the 10 % sea ice concentration contour).

Line 74-75: Coastal grid points (within 20 km of the coast) are excluded from the model performance evaluation. I didn't notice these points being masked out or flagged in some way in the forecasts released via the THREDDS server of the Norwegian Meteorological Institute. Might it be helpful to users if there is an indication of where you have confidence in the available forecast data and where users should take care.

Coastal grid points (within 20 km from the coast) are only excluded when the forecasts are evaluated using AMSR2 observations as reference, but not when the ice charts are used as reference because the ice charts are primarily based on higher-resolution satellite observations. In order to clarify this point, we have added the following sentence in section 2.2.1 describing the observations used in this study:

"Therefore, land contamination is much less present in the ice charts than in passive microwave observations, and we decided to take into account all oceanic grid points when the ice charts are used as reference (no coastal grid points excluded)."

Furthermore, while land contamination can be present in passive microwave observations and in the MET-AICE forecasts, it is not always the case. We decided to keep the coastal grid points in the forecasts delivered on the THREDDS server of the Norwegian Meteorological Institute, similarly to what is usually done for passive microwave sea ice concentration products. We do not provide any uncertainty estimates yet in the MET-AICE forecasts, but we decided to describe this issue in the paper instead.

Line 117: It isn't particularly clear how you used the datasets from 2021-2023 and why you only produced the validation on the data from April 2024 onwards. Would having a few extra years of validation assessment have made the results more robust?

We have added a new figure (see figure 4 above) showing the interannual variability in the performances of MET-AICE and ECMWF IFS between 2022 and 2024. MET-AICE has very similar performances during these three years. Furthermore we have added the following paragraph to describe this figure:

"MET-AICE and ECMWF IFS forecasts are evaluated during the period 2022 - 2024 over the full domain of MET-AICE using AMSR2 observations as reference (Fig. 4). In addition, we evaluated the MET-AICE model developed for 1-day lead time in an auto-regressive mode to predict SIC up to 10 days ahead (hereafter referred to as "MET-AICE-roll-out"). In this configuration, the SIC prediction from the previous lead time is used as a predictor to predict the next time step. While this configuration allows to improve the consistency between the time steps, it results in poorer performances than the operational version of MET-AICE. On average over all lead times, the RMSE of the SIC is about 19 % larger for MET-AICE-roll-out than for the operational version of MET-AICE, and the error for the ice edge position is about 10 % larger for MET-AICE-roll-out. Therefore, we decided to only present the results for the operational version of MET-AICE for the rest of this study. For all lead times and all the years evaluated, MET-AICE considerably outperforms persistence of AMSR2 observations (RMSE of the SIC and ice edge distance error about 28 % smaller on average), ECMWF IFS forecasts (RMSE of the SIC and ice edge distance error about 25 % and 30 % smaller, respectively), as well as ECMWF IFS bias corrected (RMSE of the SIC and ice edge distance error about 18 % and 21 % smaller, respectively). Furthermore, there is little interannual variability in the performances of MET-AICE, which suggests that re-training MET-AICE does not have to be done every year. While ECMWF IFS does not outperform persistence of AMSR2 observations for lead times up to 3 days, the bias correction allows a considerable improvement of the forecast skill for lead times up to 6 days"

**Technical corrections**

**Line 22: change "predict" to "predicts"**

We suppose that you refer to line 222 here. We replaced "predict" by "predicts" in line 222.

**Line 203: I think "less than" should be "fewer than" in this case**

"less than" has been replaced by "fewer than"

---

## Author Comment (AC2)

This is a very well-written and timely manuscript introducing MET-AICE v1.0, the first operational data-driven short-term sea ice prediction system for the European Arctic. I particularly appreciate that the evaluation is based on the most recent year of operational forecasts, which makes the results highly convincing and relevant for end-users. The manuscript is clear, the methodology is carefully described, and the system itself represents a significant step forward for operational sea ice forecasting. The authors convincingly show that MET-AICE provides skilful 10-day sea ice concentration forecasts at 5 km resolution, consistently outperforming both persistence and the Barents-2.5 km dynamical prediction system. Overall, I find the paper well-suited for publication after minor revisions. Below, I provide some suggestions that I believe could further strengthen the manuscript.

**Comments and questions**

**Prediction scheme design**

The use of separate models for each lead time is interesting. Could the authors elaborate on why this design was preferred over a more common autoregressive approach? It would strengthen the manuscript to show why this scheme provides better sea ice concentration forecasts than autoregression. Additionally, once the 10 forecasts are concatenated, do they retain physical consistency from one time step to the next?

Thanks for this comment. We agree that a discussion about that is missing in the preprint. We have added the following figure in the paper:

Figure 4. Performances of MET-AICE and ECMWF IFS during the years 2022, 2023, and 2024 over the full MET-AICE domain using AMSR2 observations as reference. MET-AICE-roll-out corresponds to using the MET-AICE model for 1-day lead time auto-regressively to predict the sea ice concentration for lead times up to 10 days. a) Root mean square error (RMSE) of the

sea ice concentration. b) Evaluation of the ice edge position (defined by the 10 % sea ice concentration contour).

And the following paragraph in the results section to discuss this:

"MET-AICE and ECMWF IFS forecasts are evaluated during the period 2022 - 2024 over the full domain of MET-AICE using AMSR2 observations as reference (Fig. 4). In addition, we evaluated the MET-AICE model developed for 1-day lead time in an auto-regressive mode to predict SIC up to 10 days ahead (hereafter referred to as "MET-AICE-roll-out"). In this configuration, the SIC prediction from the previous lead time is used as a predictor to predict the next time step. While this configuration allows to improve the consistency between the time steps, it results in poorer performances than the operational version of MET-AICE. On average over all lead times, the RMSE of the SIC is about 19 % larger for MET-AICE-roll-out than for the operational version of MET-AICE, and the error for the ice edge position is about 10 % larger for MET-AICE-roll-out. Therefore, we decided to only present the results for the operational version of MET-AICE for the rest of this study."

**Forcing model evolution**

At line 120, the manuscript notes the distribution shift in ECMWF atmospheric forecasts from one cycle to the next. To what extent do the authors envisage the need for retraining MET-AICE in the coming years as ECMWF forecasts continue to evolve? Have they evaluated model skill across past ECMWF cycles to quantify the impact?

We have added a new figure (see below) showing the interannual variability in the performances of MET-AICE and ECMWF IFS between 2022 and 2024. MET-AICE has very similar performances during these three years, suggesting that re-training the system is not necessary every year. We plan to re-train the models every time there will be modifications in the MET-AICE prediction system.

The following paragraph has been added:

"For all lead times and all the years evaluated, MET-AICE considerably outperforms persistence of AMSR2 observations (RMSE of the SIC and ice edge distance error about 28 % smaller on average), ECMWF IFS forecasts (RMSE of the SIC and ice edge distance error about 25 % and 30 % smaller, respectively), as well as ECMWF IFS bias corrected (RMSE of the SIC and ice edge distance error about 18 % and 21 % smaller, respectively). Furthermore, there is little interannual variability in the performances of MET-AICE, which suggests that re-training MET-AICE does not have to be done every year."

Furthermore, we also evaluated this in figure S3 of the supplement in Palerme et al., 2024 (<a href="https://tc.copernicus.org/articles/18/2161/2024/tc-18-2161-2024-supplement.pdf">https://tc.copernicus.org/articles/18/2161/2024/tc-18-2161-2024-supplement.pdf</a>). Note that the same neural network architecture and almost the same data (but over a different domain) were used in Palerme et al., 2024. The conclusion of this analysis was that it was beneficial to use a

longer training period (and therefore data from more ECMWF model cycles) for the sea ice concentration forecasts.

Figure 4. Performances of MET-AICE and ECMWF IFS during the years 2022, 2023, and 2024 over the full MET-AICE domain using AMSR2 observations as reference. MET-AICE-roll-out corresponds to using the MET-AICE model for 1-day lead time auto-regressively to predict the sea ice concentration for lead times up to 10 days. a) Root mean square error (RMSE) of the sea ice concentration. b) Evaluation of the ice edge position (defined by the 10 % sea ice concentration contour).

**Verification metric robustness**

How robust is the ice edge distance error to differences in smoothing across products? I would expect a sharper forecast to yield a longer ice edge, due to more small-scale information, compared to a smoother ML forecast. Clarification here would be valuable.

Thanks for this very relevant comment. We performed a sensitivity experiment to investigate this and we added a figure showing the results of this experiment in the paper. The figure and the text added in the revised version of the paper:

**In the results section:**

"In order to investigate to what extent the verification scores used in this study are influenced by the effective spatial resolution of the SIC fields, a sensitivity experiment was performed (Fig. 3). The AMSR2 SIC observations from 2013 to 2024 were used to calculate the RMSE of the SIC and the ice edge distance error for 1-day persistence of AMSR2 retrievals. The observations from the first day were smoothed using various Gaussian filters with a standard deviation ranging from 5 km (1 grid point) to 30 km (6 grid points), and compared to the original AMSR2 SIC field from the second day. Only the ice edge length from the observations of the second day at the original spatial resolution (5 km) was used to compute the ice edge distance error

regardless of the size of the Gaussian filter. In order to avoid some biases resulting from smoothing coastal grid points, the grid points closer than 50 km from the coastlines were removed from this analysis. Smoothing the AMSR2 SIC fields results in lower RMSE if the standard deviation of the Gaussian filter is between 5 and 20 km. The lowest RMSE is obtained for a Gaussian filter with a standard deviation of 10 km, with a reduction of 8.4 % compared to the RMSE obtained using the original SIC fields at 5 km resolution. This is in contrast with the ice edge distance error, which is only reduced by 0.5 % when a Gaussian filter with a standard deviation of 5 km is applied compared to the score obtained with the original resolution. Moreover, the ice edge distance error constantly increases when larger Gaussian filters are applied. Therefore, the ice edge distance error does not really favor smoother SIC fields contrary to the RMSE of the SIC."

**In the conclusion:**

"Furthermore, it is common practice to evaluate machine learning models using verification scores that are strongly correlated with the loss function. This study highlights that this can lead to spurious conclusions if no independent verification score is used in addition. Here, we show that the RMSE of the SIC can be reduced by 8.4 % due to the smoothing of the SIC fields. Therefore, we strongly recommend using more independent verification scores, such as the ice edge distance error, which does not favor smoother SIC fields as the RMSE of the SIC does."

Figure 3. Influence of the effective spatial resolution on the verification scores. The AMSR2 observations during the period 2013 - 2024 were used to calculate the RMSE of the sea ice concentration and the ice edge distance error for 1-day persistence of AMSR2 observations over the MET-AICE domain (4327 days evaluated). The AMSR2 observations from the first day were smoothed using various Gaussian filters with a standard deviation ranging from 5 km (1 grid point) to 30 km (6 grid points), while the original AMSR2 observations from the next day were used as reference for computing the scores. The ice edge distance error was computed using the ice edge length from the observations of the second day at the original spatial resolution (5 km). The grid points closer than 50 km from the coastlines were excluded from this analysis. a) The mean ratio between the scores after smoothing and the scores using the original data (a score lower than 1 means that the smoothing results in a better score). The shaded areas represent the standard deviations of the verification scores. b) The fraction of scores that are better after smoothing the AMSR2 sea ice concentration fields. c) AMSR2 sea ice concentration observations at the original spatial resolution (5 km) on 01/02/2025. d, e, f) AMSR2 sea ice concentration observations on 01/02/2025 smoothed using Gaussian filters with a standard deviation of 5 km (d), 15 km (e), and 25 km (f). The scores on the top left corners of the maps show the RMSE of the sea ice concentration (top, in %) and the ice edge distance error (bottom, in meters) for the comparison between the AMSR2 observations from 01/02/2025 and 02/02/2025

**Sensitivity to different forcings**

Have the authors tested MET-AICE under different atmospheric forcings than those used in training (e.g., AROME-Arctic)? Would the model's forecast skill improve or deteriorate in such cases, as seen in Barents-2.5?

We have not tested running MET-AICE under different atmospheric forcings, but it is likely that the forecast skill would deteriorate if another forcing (such as AROME-Arctic) would be used. A machine learning model is trained to learn the correlation (non linear) between features of a dataset (the predictors) and the target variables. When a dataset with different properties than the one used for training is used as a predictor, some biases can be expected in the predictions. For example, AROME-Arctic has a much higher spatial resolution (2.5 km) than ECMWF IFS (9 km). Since MET-AICE has been trained using ECMWF IFS forecasts, MET-AICE has learnt some convolutional kernels (3 x 3 grid points) at maximum 9 km resolution, and should not be able to take advantage of the higher spatial resolution of AROME-Arctic forecasts. It is also worth noting that AROME-Arctic does not cover the full domain of MET-AICE, and that convolutional neural networks are not designed to work on a grid with a varying resolution. For grids with varying resolutions, graph neural networks would be much more relevant than convolutional neural networks. Therefore, we did not include any experiment like this in the revised version of the paper.

Relatedly, it would be instructive to assess MET-AICE in controlled "idealised" experiments not present in the training data (e.g., constant zonal winds, uniformly melting temperatures across the domain). Do the forecasts behave in line with expected sea ice physics?

We have investigated this using either a constant wind field of about 10 meter per second with a north west direction (blowing towards the south east) or a constant 2-meter temperature field of 280 kelvins. The forecasts were evaluated during the period April 2024 to March 2025. On average, the forecasts with a constant wind field produced an increase of the sea ice extent as expected, and the forecasts with a constant temperature field of 280 K produced a decrease of the sea ice extent as expected. However, these results were not systematic. In particular for the constant wind field, an increase of the sea ice extent was observed in only 65 % of the cases, and there was a lot of spatial variability in these results. These results are difficult to interpret because of this variability and because of the lack of ground truth.

Furthermore, convolutional neural networks are trained to recognize spatial patterns through the convolution operations. On constant fields, the neural networks would not recognize any spatial patterns learnt during training, which would likely result in inaccurate predictions. Moreover, providing a dataset to a supervised neural network which is very different from any sample used for training the neural network would also likely result in inaccurate predictions. Therefore, we did not include any experiment like this in the revised version of the paper.

Along these lines, has the system been tested using ECMWF ensemble members in addition to the control forecast? Does MET-AICE produce a reasonable spread in SIC forecasts under such perturbations?

Thanks for this comment. We have investigated this in the revised version of the paper. Please find below the figure and the text added in the revised version of the paper:

Figure 10. Evaluation of probabilistic sea ice forecasts. The forecasts starting on Mondays, Wednesdays, and Fridays from April 2024 to March 2025 are evaluated in the shared domain between MET-AICE and Barents-2.5. a, b, c) Comparison between the sea ice probability (probability that the sea ice concentration exceeds 10 %) and the observed frequency of occurrence in AMSR2 observations. d) Probabilistic ice edge distance error (ratio of the Spatial Probability Score over the observed ice edge length) depending on the lead time using AMSR2 observations as reference. e, f, g) Comparison between the sea ice probability and the observed frequency of occurrence in the ice charts. h) Probabilistic ice edge distance error depending on the lead time using the ice charts as reference.

In the Dynamical models section (section 2.2.2):

"This study primarily focuses on evaluating the MET-AICE operational forecasts that are deterministic and have daily time steps. Therefore, the daily means of SIC fields from the dynamical models are evaluated. These datasets consist of the high-resolution ECMWF IFS (HRES) forecasts, the TOPAZ5 ensemble mean, the Barents-2.5 member forced by AROME-Arctic, and the mean of Barents-2.5 members forced by ECMWF-ENS forecasts. In

addition, we also investigated producing probabilistic forecasts with MET-AICE in this study. For this experiment, MET-AICE forecasts were compared to the first 10 ensemble members from ECMWF IFS (ENS), the 10 TOPAZ5 ensemble members, and the 5 Barents-2.5 ensemble members forced by ECMWF-ENS produced at 00:00 UTC."

In the verification section (section 3.2):

"While deterministic SIC forecasts are evaluated in most of this study, an experiment was performed with probabilistic forecasts. The ice edge position in probabilistic forecasts is assessed using the sea ice probability ( $SIP_{forecasts}$  in equation 2), which is defined as the fraction of ensemble members with a SIC higher or equal to 10 %, whereas a binary sea ice probability is used for the observations ( $SIP_{observations}$  in equation 2). Then, the forecasts are evaluated using the ratio of the Spatial Probability Score (SPS, Goessling and Jung, 2018) over the ice edge length in the observations used as reference (hereafter referred to as "probabilistic ice edge distance error", equation 2). This metric was introduced by Palerme et al. (2019) and can be considered as the probabilistic extension of the ice edge distance error."

**In the results section:**

"In addition to evaluating the operational MET-AICE deterministic forecasts, we investigated if we could use several ensemble members from ECMWF IFS (ECMWF-ENS) for producing probabilistic SIC forecasts with MET-AICE. We used the first 10 ensemble members of ECMWF-ENS for the atmospheric predictors (10-meter wind and 2-meter temperature) to produce a set of 10 ensemble members with MET-AICE. In Fig. 10, the forecasts starting on Mondays, Wednesdays, and Fridays from April 2024 to March 2025 are evaluated in the shared domain between MET-AICE and Barents-2.5 in order to include the three dynamical models. Overall, MET-AICE and the three dynamical models produce ensemble forecasts that are overconfident (not enough ensemble spread), which means that low sea ice probabilities (SIP) are observed more frequently than predicted whereas high SIP are observed less frequently than predicted. While the MET-AICE forecasts are particularly overconfident for 1-day lead time, they have a similar spread as the TOPAZ5 forecasts for 10-day lead time. Among the prediction systems, ECMWF IFS produces the largest ensemble spread. Furthermore, the probabilistic ice edge distance error is lower in MET-AICE forecasts than in all dynamical models, except for lead times of 9 and 10 days when the ice charts are used as reference."

**Missing Literature**

I recommend citing also these papers: https://doi.org/10.3389/fmars.2023.1260047 and https://doi.org/10.1029/2024JH000433. Even though the latter does not pertain to the Arctic, it's still a good example of a data-driven prediction system targeting sea ice.

We have included these two references in the following sentence of the introduction:

"Several studies have recently shown that data-driven sea ice forecasting systems trained on satellite observations can be skillful (e.g. Grigoryev et al., 2022; Ren et al., 2022; Chen et al., 2023; Keller et al., 2023; Lin et al., 2023; Koo and Rahnemoonfar, 2024), and can provide more accurate forecasts than dynamical models (Andersson et al., 2021; Palerme et al., 2024; Kvanum et al., 2025; Lin et al., 2025). "

Citation: https://doi.org/10.5194/egusphere-2025-2001-RC2